# What the eastern African stone tool evidence tells us about Plio-Pleistocene hominin extinctions

## Research Article

Plio-Pleistocene; eastern Africa; *Australopithecus*; *Paranthropus*; early *Homo*; *Homo ergaster/erectus*; Stoneworking modes A-I

**Corresponding author:**
John Shea;
Email: john.shea@stonybrook.edu

## John J. Shea [ID]

Stony Brook University, New York, USA

### Abstract

This paper examines the stone tool evidence associated with extinctions among *Australopithecus*, *Paranthropus*, and *Homo* in Eastern Africa between 0.8 and 3.5 Ma. It does this using Stoneworking Modes A-I, a relatively new framework for comparing stone tool evidence, and data from the Eastern African Prehistoric Stoneworking Survey. While the evolutionary transition between early *Homo* and *H. ergaster/erectus* seems correlated with a shift from occasional to habitual stoneworking, *Australopithecus'* and *Paranthropus'* last appearance dates do not appear correlated with major changes in the archaeological record.

### Impact statement

Traces of percussive stoneworking follow the origin, evolution, and dispersals of hominins in the Genus *Homo* for more than 2.5 million years. Archaeologists commonly ask questions about how stoneworking and stone tools contributed to ours and earlier hominins' evolutionary success. This paper tacks differently. It asks what change and variability in the Eastern African Plio-Pleistocene archaeological record dating between 0.8 and 3.5 million years ago tell us about hominin extinctions. Changes in stoneworking accompany evolutionary changes in the Genus *Homo*, but not changes in now-extinct hominin genera, *Australopithecus* and *Paranthropus*.

### Introduction

Humans have contemplated our extinction for a very long time (Lynskey, 2025). One thinks this concern ironic, for our global geographic distribution and large population (extraordinary for creatures our size) make us immune to all but a handful of biosphere-level extinction threats (Shea, 2023). Thinking about Pleistocene hominin extinctions takes us to a "lost world," in which long-term hominin survival was far from a certain thing.

With respect to extinctions, a curious pattern stalks the hominin fossil record. First appearance dates (FADs) of our species (*Homo sapiens*) correlate with last appearance dates (LADs) of other hominins. There are a few exceptions to this pattern, such as in the East Mediterranean Levant (Shea, 2025), but generalizing broadly, humans' arrival anywhere is bad news for large mammals, including other hominins, everywhere (Martin, 1989). As Marean (2015) puts it, we are "the most invasive species of all." But does this pattern arise from something uniquely human, or was it established early on in the evolution of the Genus *Homo*? To answer this question, we must examine later Pliocene and early Pleistocene evidence from Eastern Africa (Ethiopia, Kenya, Tanzania, and adjacent countries). There, between 0.8 and 3.5 Ma, the earliest species of our Genus, *Homo*, differentiated themselves from other hominins and began their global dispersal (Fleagle et al., 2010). For brevity's sake, this paper uses the term, "Plio-Pleistocene," for this period.

This paper uses Stoneworking Modes A-I, a diagnostic framework developed for comparing stone tool evidence across different archaeological age-stages and research traditions (Shea, 2013). One of the main obstacles for comparing Plio-Pleistocene stone tool evidence (or the evidence from any period, really) is that archaeologists organize their observations in so diverse ways that comparisons inevitably require one to resort to "lowest common denominator" descriptions of stone tools. Eastern Africa suffers acutely from such "lithics systematics anarchy" due to the variety of imported and indigenous research traditions in place among archaeologists working in that region (Shea, 2023). More detailed comparisons require an archaeologist to re-examine artifacts their colleagues have already published, a task for which it can be difficult to obtain funding, especially for recently-published evidence. Stoneworking Modes A-I do not require this. One can usually ascertain from published literature for which among Stoneworking Modes A-I an archaeological lithic assemblage preserves evidence.

Using Stoneworking Modes A-I does not preclude using more conventional artifact-focused approaches. If all one needs to know is whether Archaeologist A and Archaeologist B are writing

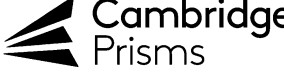

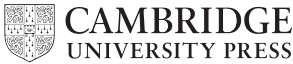

about the same or different things when they attribute specific sets of stone tools to one or another stone tool industry or some other higher-order assemblage-groups, then Stoneworking Modes A-I can allow one to quickly discover and investigate any differences in their attributions. Such differences can be resolved quickly either by communications with colleagues or by targeted physical examination of the artifact collections in question. If one's research question requires one know how many choppers vs. scrapers are present or parametric measurements of specific artifacts, then Stoneworking Modes A-I poses no obstacles to doing so. (To invoke a medical analogy, diagnosing a patient as having the flu does not preclude recording their sex or their weight, but it speeds the path to treating them.) Used in this way, Stoneworking Modes A-I can be an aid to collaboration and to replicability in archaeological stone tool analysis.

Using Stoneworking Modes A-I also allows more inclusive comparisons than otherwise. It has proven useful for testing hypotheses about behavioral differences among "technological primates" (Shea, 2017b), for synthesizing the stone tool evidence for the full sweep of Eastern African prehistory (Shea, 2020), and for documenting changes in human (*H. sapiens*) problem-solving strategies over the course of our global dispersal (Shea, 2023). Researchers have also used it to test hypotheses about "cumulative culture" and for other purposes (Paige and Perreault, 2022, 2024).

One did not develop Stoneworking Modes A-I specifically to investigate Plio-Pleistocene hominin extinctions; but if it proves useful in this task, then colleagues may find Stoneworking Modes A-I helpful in researching hominin extinctions in other regions and time periods. By "useful" one means that it achieves either the same or additional insights into those extinctions than those archaeologists have thus far achieved using other approaches to the stone tool evidence. If, on the other hand, applying Stoneworking Modes A-I to the Eastern African Plio-Pleistocene lithic evidence produces "data anarchy" offering up no insights whatsoever, then it will have failed.

Finally, and if only to manage expectations, one needs to state what this paper does not do. It does not review of the different ways in which researchers organize the Plio-Pleistocene stone tool evidence from Eastern Africa (e.g., ; Leakey, 1971; Isaac et al., 1997; Mora Tocal et al., 1992; de la Torre et al., 2003; de la Torre, 2004; de la Torre and Mora, 2005; Delagnes and Roche, 2005; Diez-Martin et al., 2010; Stout et al., 2010; Roche et al., 2018; Shea, 2020; Mesfin et al., 2026) or the Pleistocene lithic evidence from elsewhere (Bordes, 1961; Laplace, 1974; Debénath and Dibble, 1994; Tostevin, 2012; Boëda and Chazan, 2023). Doing justice to this topic would vastly exceed space available and distract from its main concern, Plio-Pleistocene hominin extinctions.

This paper neither discusses nor critiques the higher-order groupings of stone tools (named industries, traditions, technological complexes, etc.) currently in use in Eastern African "deep-time" prehistory (see Shea, 2020). Using Stoneworking Modes A-I does not require one to organize the stone tool evidence in terms of these entities.

## Methods and materials

### Hominin fossils

Paleontologists group Plio-Pleistocene hominin fossils into at least three major genera, *Australopithecus*, *Paranthropus*, and *Homo*. The principal quality that recommends all three genera as possible stoneworkers is their loss of their Miocene ancestors' large self-sharpening canine teeth (Fleagle et al., 2026). Stone tools are essentially artificial teeth (Shea, 2017b), and so, all of these hominins had possible motives for using stone cutting tools, either naturally-fractured rocks or, if and when such stones were unavailable, the edges of freshly-broken bone (Eren et al., 2025). As to means and opportunity, fossils of all three genera occur in places where rocks amenable to percussive fracturing are available and where fractured fossil bones appear. All appear to have had opposable thumbs and capacities for power and precision grasping, albeit with some variation in capability (Kivell et al., 2023). We can rule out none of them as possible stone tool makers and users (Prat, 2025).

All of these creatures appear to have been sexually dimorphic (Fleagle et al., 2026), and as such the ranges concerning their inferred height and weight express lower values for presumed females and higher ones for males. Estimates quoted here combine those recently published by Monson et al. (2026) with those from the website of the United States National Museum of Natural History (Smithsonian Institution) Human Evolution Research Program (https://humanorigins.si.edu/).

*Australopithecus afarensis* is the best-documented among Eastern Africa's several named Plio-Pleistocene australopithecines (Hammond and Mongle, 2023; Fleagle et al., 2026). These hominins stood about 105–151cm tall and weighed between 29 and 42 kg. Their brain volume was around 500 cm$^3$, not much larger than that of chimpanzees, but their molar teeth are larger than those of chimpanzees and of living humans. Their post-cranial remains and footprints attributed to them at Laetoli in Tanzania show they walked bipedally, but with some differences from living human bipedalism (Raichlen et al., 2010).

*Paranthropus* comprises mainly *P. boisei* and *P. aethiopicus* (Hammond and Mongle, 2023; Fleagle et al., 2026). These hominins stood about 124–137cm tall and weighed between 34 and 49 kg. Their brain volume, 400–550 cm$^3$, was the same or a little larger than that of *Australopithecus*, but smaller than those of Early *Homo* and *H. ergaster/erectus*. *Paranthropines'* most distinctive features involve their powerful biting and chewing capacities. These include very large molars, wide zygomatic arches and a sagittal crest that anchored enlarged temporalis and masseter muscles. Postcranial remains suggest a body shape-more-or-less like *Australopithecus* -long arms, short legs, and a flaring ribcage.

Fossils of the Genus *Homo* differ from australopithecines and paranthropines in featuring larger brains, smaller molars, reduced prognathism, and reduced anatomical supports for chewing. Paleontologists recognize major differences between Early *Homo* and *Homo ergaster/erectus*. Views diverge over which Eastern African fossils are *H. ergaster* vs. *H. erectus* (Klein, 2009; Fleagle et al., 2026); so, here we treat them together.

Minimally, Early *Homo* includes *Hesydrus habilis* and (arguably) *H. rudolfensis* (Hammond and Mongle, 2023; Fleagle et al., 2026). These creatures' height ranged from 100 to 130 cm. Estimates of their brain size vary widely, but most fall between 500 and 800 cm$^3$. Post-cranially, they had long arms, short legs and other features suggesting that, while they show distinctive features of terrestrial locomotion, they retained ancestral hominin skeletal adaptations to arboreality (climbing in trees) (Grine et al., 2026).

*Homo ergaster/erectus* had larger brains than Early *Homo* (750–1,000 cm$^3$) and they were taller (160–189 cm). Post-cranially, *Homo ergaster/erectus* had a more derived postcranial skeleton, one with hallmarks of "endurance bipedalism" similar to those seen among younger hominins (Lieberman et al., 2008). These features include a long and narrow foot with an enlarged hallux aligned parallel to the other toes, relatively long legs and relatively short arms, and a rounded ribcage with a true waist (a large gap between lower ribs

and ilium) (Walker and Leakey, 1993; Fleagle and Lieberman, 2015; Hammond and Mongle, 2023; Fleagle et al., 2026).

## Stone tools

Why did Plio-Pleistocene hominins make stone tools? The archaeological consensus holds that they did so as aids to pre-oral food-processing, and perhaps to make cutting tools to shape other implements out of wood. Stone tool cut-marks on bone indicate that some stone tools were used as aids to butchery (Potts and Shipman, 1981; Bunn et al., 1986; Blumenschine and Pobiner, 2007), and lithic wear-trace analyses hint at other possible uses on wood and softer plant matter (Keeley and Toth, 1981; Lemorini et al., 2014). Eastern Africa lacks preserved wooden artifacts of this age, though some are known from sites in other regions, such as Gesher Benot Ya'acov in Israel (Goren-Inbar et al., 2002). Oddly, and in contrast with the stone tools recent humans use, many Plio-Pleistocene stone tools bear traces of extensive percussion that did not result in large-scale fracturing but rather "comminution" (multiple overlapping and incompletely-propagated fractures) (Mora and de la Torre, 2005; Diez-Martín et al., 2009). The nature of these percussive tasks remains unknown. We can say little about them other than that they created noise. Indeed, we should not dismiss the possibility that noise-making for some social and/or signaling purpose may have motivated Plio-Pleistocene hominins' percussive stoneworking.

Archaeologists' stone artifact typologies and higher-order systematics for the Eastern African Plio-Pleistocene stone tool evidence vary historically and between different (mostly European) research traditions (Shea, 2020). This variation makes large-scale comparisons difficult. I developed Stoneworking Modes A-I to enable comparisons of the stone tool evidence without having to disentangle that evidence from these pre-existing archaeological systematics, either in Eastern Africa or anywhere else (Shea, 2013). Modes A-I is not an artifact typology. It is a standardized set of diagnoses of stoneworking methods. It comprises nine modes, some of which it further divides into submodes, distinct variants of the major modes recognized across research traditions. Specific sorts of artifacts indicate each of these modes and submodes, but some artifacts can indicate more than one mode/submode and vice versa. For this reason, Stoneworking Modes A-I registers presence/absence only (Shea, 2013, 2017a, 2017b, 2020, 2023). It does not register differences in lithic raw materials (though one could do so if one wanted to), and it does not take into account archaeologists' (and others') speculations about stone tool functions.

This paper uses data from the Eastern African Prehistoric Stone-working Survey (EAPSS) (Shea, 2020), a database compiling observations about Stoneworking Modes A-I on more than 250 lithic samples dating from the Later Pliocene to the Iron Age. This database, updated regularly, is available at this website: https://sites.google.com/a/stonybrook.edu/john-j-shea/eastern-african-stone-tools-east-typology?authuser=0.

Eastern African Plio-Pleistocene assemblages do not preserve all of Stoneworking Modes A-I. The sections below describe those the EAPPS identifies as represented among in them (see also Figure 1).

One assumes most readers know basic terms for stone tools, but for any who do not, a simple description follows: Percussive stone-working involves striking one rock (a core) with another one (a hammerstone) or some other hard object until fracture detaches a large portion of that rock (a flake). "Hierarchy" refers to core reduction in which flakes detached from one side of an edge differ from those detached from the opposite side of that edge. "Retouch" describes smaller-scale fractures initiated on core or flake edges. Archaeologists use many terms for retouched artifacts, but the most basic of these and those relevant for Plio-Pleistocene archaeology are core-tools and retouched pieces/retouched flakes. For more detailed descriptions of stoneworking, techniques, methods, and terms for its products, see Whittaker (1994) and/or Shea (2020).

Mode A, *anvil percussion*, involves throwing or striking a core against another rock or some other hard substrate, such as a bedrock exposure in such a way that the resulting damage to the core includes either a few large fractures or multiple incompletely-propagated fractures, "stacks" of short step-terminated fractures, and/or comminution (Shea, 2020). Diagnostic artifacts include stone percussors, hammerstones, spheroids and subspheroids.

Mode B, *bipolar core reduction*, creates fractures by striking the uppermost surface of a core, flake, or some other rock that rests on a hard substrate. Diagnostic artifacts include bipolar cores and scaled pieces. Some artifacts damaged through so-called "passive-rest" percussion may simultaneously fit the criteria for Mode B and Submode G1 (discussed below).

Mode C, *short non-hierarchical core reduction*, involves sequential and relatively large (>2cm long) non-hierarchical flake removals from clasts (rounded rocks) or angular rock fragments (both of whose cardinal dimensions [length, width, and thickness] are roughly equal to one another). Diagnostic artifacts include bifacial choppers, discoids, and polyhedrons.

Mode D, *flake retouch*, detaches contiguous series of relatively small (10–20 mm long) flakes from the edge of a flake or flake fragment. Such retouch can be either hierarchical or non-hierarchical. Characteristic artifacts include retouched pieces.

Submode D1, *scraper retouch*, removes a continuous series of flakes at right angles to the edge of a flake/flake fragment, thereby creating an edge that is relatively acute in cross-section. Diagnostic artifacts include scrapers, notches, and denticulates.

Submode D2, *macrolithic backing/truncation*, creates relatively steep (>70°) retouch on artifacts longer than 30mm. Characteristic artifacts include backed and/or truncated pieces.

Submode D4, *burination*, propagates a fracture roughly perpendicular to a flake's dorsal or ventral surface. Diagnostic artifacts include burins, burin flakes, and tranchet flakes. Burin flakes grade into elongated flakes struck from flake-cores (discussed below).

Submode D5, *convergent retouch*, brings together retouched edges that intersect at acute angles in both plan and profile views. Diagnostic artifacts include points, awls, and convergent scrapers.

Submode D7, *flake-core reduction*, detaches relatively large flakes (>20 mm long) from flake fragments. Diagnostic artifacts include cores-on flakes and so called "Kombewa flakes," flakes that propagated under another flake's bulbar eminence and thus appear to have two ventral surfaces.

Mode E, *elongated non-hierarchical core reduction*, involves non-hierarchical flake detachments from an elongated piece of stone (a core, flake, or tabular rock fragment) in such a way that flakes propagate roughly perpendicularly to the core's long axis.

Submode E1, *long core-tool (LCT) production*, creates bifacial cores with width/ thickness ratios usually greater than 3 to 1. Diagnostic artifacts include handaxes, cleavers, picks, bifaces, "Acheulean elements," and other so-called "large cutting tools."

(Modes A-I uses the term, "long-core tools" rather than "large cutting tools" because it accurately describes the artifacts in question without exaggerating their size or making assumptions about their functions. Most of these LCTs are not much larger than a human hand (Marshall et al., 2003) or a smart phone (iPhone 17™=149.6 × 71.5 × 7.95 mm), and only a small number (dozens

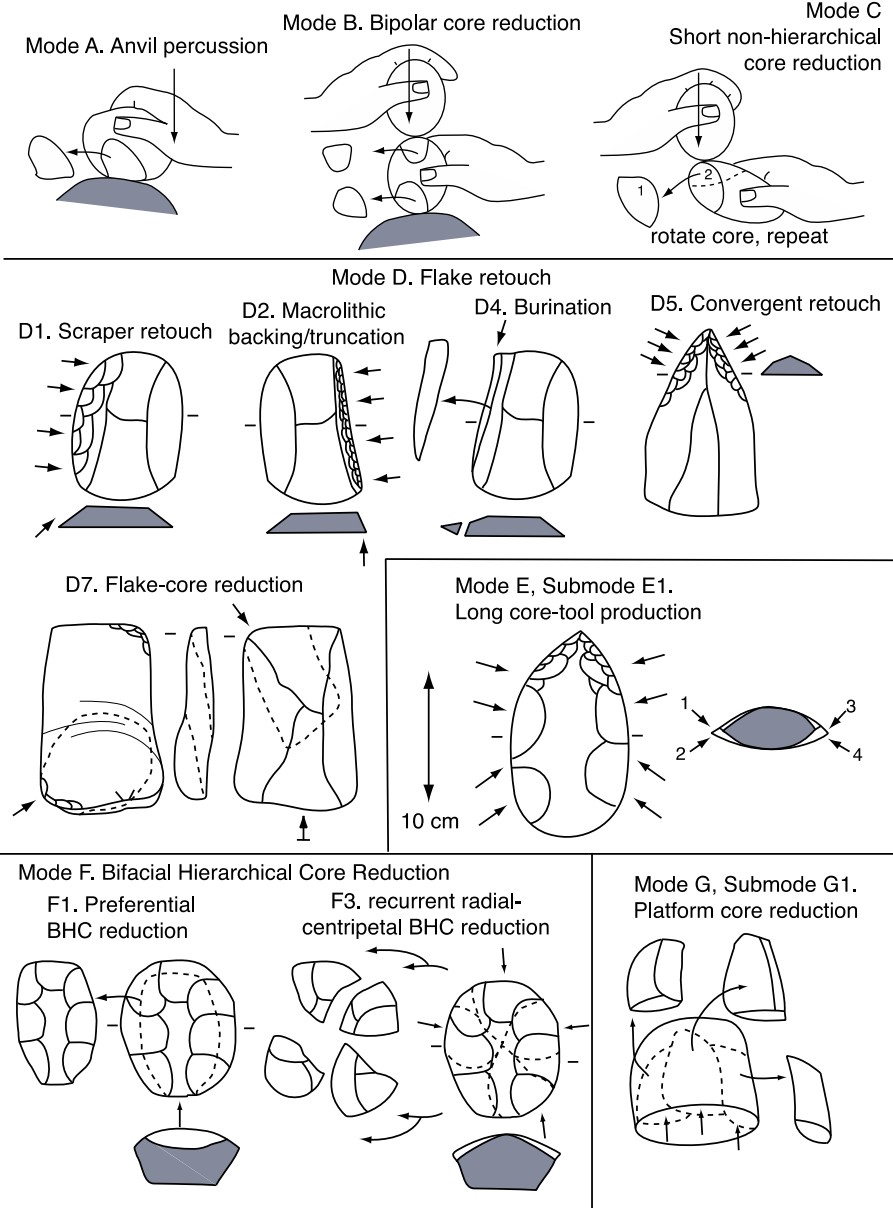

**Figure 1.** Schematic diagram of the Stoneworking Modes represented among Eastern African Plio-Pleistocene assemblages.

out of the thousands held in archaeological museums) preserve evidence of having been used to cut anything (Shea, 2017b).)

Mode F, *bifacial hierarchical core (BHC) reduction*, detaches relatively short flakes from one side of a worked edge and longer flakes from the other side of that same edge. Ridges separating the shorter fracture scars ("faceting") create a convexity that concentrates percussor force onto a small area, propelling fracture propagations initiated there farther than would otherwise be the case. Levallois cores are BHCs, but Mode F also includes BHCs that some archaeologists might not classify as Levallois cores.

*Preferential BHC reduction* (Submode F1) detaches a single relatively large flake from one side of the core's working edge.

In *recurrent radial-centripetal BHC reduction* (Submode F3), an overlapping series of flake scars converge with one another from multiple points around the core's circumference.

Mode G, *unifacial hierarchical core (UHC) reduction*, entails a stable hierarchy of fracture initiation and fracture propagation surfaces in which the fracture initiation surface is roughly planar and maintained at nearly a right angle to the curved and convex fracture propagation surface. Most Plio-Pleistocene instances of Mode G are *platform core reduction* (Submode G1), in which stoneworkers detach relatively short flakes (length = <2×width) from tabular or roughly hemispherical cores. Diagnostic artifacts include unifacial choppers and core-scrapers.

## Results and analysis

Viewed in terms of Stoneworking Modes (see Table 1), the Eastern African Plio-Pleistocene stone tool evidence features two major inflection points, one around 3.4 Ma, the other around 1.9 Ma.

The period around 3.4 Ma marks the earliest evidence of occasional stoneworking. The stone tool evidence from this date onwards to 1.9 Ma features evidence of Modes A, B, C, and G1, *anvil percussion, bipolar core reduction, short non-hierarchical core*

**Table 1.** Plio-Pleistocene Stone tool assemblages from the Eastern African prehistoric stoneworking survey

| Sample | Younger date | Older date | A | B | C | D1 | D2 | D4 | D5 | D7 | E1 | F | F1 | F3 | G1 | Notes | References |
|---|---|---|---|---|---|---|---|---|---|---|---|---|---|---|---|---|---|
| West Turkana Lomekwi 3 | 3,300 | 3,300 | + | + | + | | | | | | | | | | + | | 1 |
| Hadar, Afar (AL 894) | 2,600 | 2,600 | | | + | | | | | | | | | | | | 2 |
| Ledi-Geraru, Bokol Dora 1 Locality | 2,580 | 2,610 | + | | + | | | | | | | | | | + | | 3 |
| Gona (Busidima Formation) | 2,500 | 2,600 | | | + | + | | | | | | | | | + | | 4 |
| Gona (Afar Dist.) | 2,300 | 2,600 | | | + | | | | | | | | | | + | 1 | 5 |
| Hadar (Afar Dist.) | 2,300 | 2,300 | | | + | | | | | | | | | | + | 2 | 6 |
| West Turkana Lokalalei 1 | 2,300 | 2,400 | | | + | | | | | | | | | | | | 7 |
| West Turkana Lokalalei 2C | 2,300 | 2,400 | + | + | + | + | + | | | | | | | | + | | 8 |
| Omo Shungura Formation Member F | 2,200 | 2,400 | + | + | + | | | | | | | | | | | 3 | 9 |
| Kanjera | 2000 | 2000 | + | | + | + | + | | | | | | | | | 4 | 10 |
| Kanjera South | 2000 | 2000 | | | + | + | | | | + | | | + | + | | | 11 |
| Fejej FJ-Ia | 1900 | 1900 | + | + | + | + | | | | | | | | | + | | 12 |
| Olduvai Gorge Bed I & Lower Bed II | 1860 | 1860 | + | + | + | + | ? | + | + | | + | | | | + | 5 | 13 |
| West Turkana Kokiselei 1, 5–6 | 1760 | 1760 | | | + | | | | | | | | | | | | 14 |
| West Turkana Kokiselei 4 | 1760 | 1760 | | | + | | | | | | + | | | | | | 15 |
| West Turkana Naiyena Engol 2 | 1700 | 1800 | | + | + | | | | | | | + | + | | | | 16 |
| Melka Kunture >1.6 Ma | 1,600 | 2000 | + | + | + | + | + | | + | | + | | | | + | 6 | 17 |
| East Turkana/Koobi Fora Formation, KBS Member Burgi Fmn. | 1,600 | 1800 | + | + | + | + | + | | | | | | | | + | 7 | 18 |
| Konso-Gardula KGA6-A1 Locus C, | 1,600 | 1800 | | | | | | | | | + | | | | | | 19 |
| Nyabosusi NY 18 | 1,500 | 2000 | | | + | + | | | | | | + | + | | | | 20 |
| Chesowanja Chemoigut Fmn. | 1,400 | 1,600 | | | + | + | | | | | | | | | | | 21 |
| Olduvai mid-upper bed II | 1,200 | 1,600 | + | + | + | + | ? | + | + | | + | | | | + | **8** | 22 |
| Peninj, EN & ES sites | 1,200 | 1,500 | + | | + | ? | | | | | + | + | + | | | | 23 |
| Peninj, ST Site Complex | 1,200 | 1,500 | + | | + | + | | | + | | | + | + | + | + | **9** | 24 |
| Konso-Gardula | 1,000 | 1,600 | | | | | | | | | + | | | | | 10 | 25 |
| Middle Awash Bouri A1 (BOU-A1) | 1,000 | 1,000 | | | | | | | | | + | | | | | | 26 |
| Isenya Units V & VI | 900 | 1,000 | + | | + | | | | | | + | | + | | | | 27 |
| Kariandusi | 900 | 1,100 | + | | | | | | | | + | | | | | | 28 |
| East Turkana, Koobi Fora Formation, Okote member | 800 | 1,600 | + | + | + | + | + | | + | | + | | | | + | **11** | 29 |

*Notes*: Specific sites included: 1. OGS-6, OGS-7, EG-10, EG-12, 2. AL-666, AL-894, 3. FtJi 1, 2, 5, Omo 57, 123, 4. KS1, KS2, 5. DK, FLK NN levels 1–4, FLK Zinj & upper levels, FLK N levels 1–6, Deinotherium level, HWK East, 6. Karre 1, Gombore I Level B, Iy, Garba IV Levels C-D, 7. Fxjj 1–3, 4–9, 13, 14, 8. HWK East: Sandy Conglomerate Levels 3–5, FLK North Sandy Conglomerate, MNK Skull Site, EFHR, MNK Main Occupation, FC West, FC, SHK, TK, BK, 9. Omo 57, 123, 10. KGA 4-A2, KGA 10-A11, KGA7-A1, A2, A3, KGA12-A1, KGA20-A1, A2, 11. FwJj1, GaJi 5, FxJj 11, 15–18, 20, 23, 37–38, 50, 63–64.

References: 1. Harmand et al. (2015), 2. Hovers (2009), 3. Braun et al. (2019), 4. Semaw et al. (2009), 5. Semaw et al. (2009), 6. Kimbel et al. (1996), (Goldman-Neuman and Hovers, 2012), 7. Kibunjia (1994), 8. Prat et al. (2005), 9. Merrick and Merrick (1976), Howell et al. (1987), Delagnes et al. (2011), de la Torre (2004), 10. Plummer et al. (1999), 11. Bishop et al. (2006), 12. Asfaw et al. (1991), Barsky et al. (2011), 13. Leakey (1971), de la Torre and Mora (Delagnes and Roche, 2005), 14. Lepre et al. (2011), 15. Lepre et al. (2011), 16. Roche et al. (2018), 17. Chavaillon and Piperno (2004), 18. Isaac and Isaac (1997), 19. Beyene et al. (2013), 20. Texier (1995), 21. (Gowlett et al., 1981), 22. Leakey (1971), de la Torre and Mora (Delagnes and Roche, 2005), 23. Domínguez-Rodrigo et al. (2009), 24. de la Torre and Mora (Diez-Martín et al., 2009), 25. Beyene et al. (2013), 26. Schick and Toth (2017), 27. Roche et al. (1988), 28. Shipton (2011), 29. Isaac and Isaac (1997).

*reduction*, and *platform core reduction* (respectively). After 1.9 Ma evidence appears for Stoneworking Modes D, E, and F, *flake retouch, LCT production, and BHC reduction.*

The stoneworking methods present before 1.9 Ma are, to put it simply, simple. They work on pretty much any rock with even an approximation of conchoidal fracture. That they appear among artifacts made by novice craft/hobby knappers and even persons with little or no prior experience or knowledge about stone tools (Shea, 2015; Snyder et al., 2022) suggests one can explain their occurrences in terms of improvisation involving local responses to immediate needs. Some teaching and learning and carrying rocks from one place to another probably went on among the Plio-Pleistocene stoneworkers who made these artifacts, much as it appears to do among living primates that use stone tools (McGrew, 1992; Haslam et al., 2017). (One thinks it would be odd indeed if it did not do so.)

How occasional was occasional? This earliest phase of Eastern Africa's lithic record boasts an impressive gap nearly a million years

long between the Lomekwi 3 evidence and the next dated appearances of lithic artifacts around 2.5–2.6 Ma, such as Nyayanga and Namorotukanan (both in Kenya) (Plummer et al., 2023; Braun et al., 2025). Even after stoneworking seems to begin in earnest after 2.5 Ma, many other long gaps appear in the evidence from the same sedimentary basins. If Eastern African Plio-Pleistocene hominins were as profligate stoneworkers as their Holocene human descendants, then there would still be gaps, but many more of them and of shorter duration.

The period after 1.9 Ma appears to signal a shift to increasingly habitual stoneworking. To be clear, I am not arguing that occasional and improvisational stoneworking stopped, but rather that habitual stoneworking augmented it. Modes D, E, and F all require more time, impulse control, planning, and arguably expertise than Modes A, B, C, and G1 (Key et al., 2026). Diagnostic artifacts for Modes D, E, and F rarely appear among stone tools novice flintknappers produce spontaneously (i.e., without expert tutoring) for the simple reason one can easily make consequential errors in creating them (Shea, 2015).

Carefully retouching the edge of a flake can restore a use-dulled edge to functionality, but doing so haphazardly or wrongly can diminish that edge's cutting potential. For example, retouch initiated by striking along the edge of a flake's smooth ventral surface "follows" that surface's contour, creating a straight or slightly curved edge that cuts well. Striking both dorsal and ventral sides creates jagged edges that exert greater amounts of drag while cutting. (Picture the edge of a woodcarving knife vs. that of a cross-cut saw.)

Imposing an elongated shape on a core, such as an LCT, makes that core vulnerable to lateral fracturing and end-shock in ways that detaching flakes from shorter non-hierarchical cores does not (Callahan, 1979; Whittaker, 1994; Patten, 2009). Experiments reproducing LCTs suggest novices and relatively inexperienced stoneworkers can produce such artifacts, but they also show that experienced stoneworkers do so more swiftly and efficiently (i.e., detaching fewer flakes in the process and with fewer catastrophic errors) (Key et al., 2026).

Bifacial hierarchical core reduction (Mode E) requires a stoneworker to maintain a stable hierarchy between the side of an edge that serves as a source of fracture initiation points and the other side from which they detach longer, wider, and thinner flakes (Boëda, 1995).

These observations suggest that after 1.9 Ma, some hominins were devoting considerably more than the minimal amounts of thought, time, and labor to stoneworking. Percussive stoneworking also entails an imprecisely quantifiable risk of injury (Gala et al., 2023). These increased costs and risks only make sense if hominins were engaging in more habitual stone tool use, using stone tools for longer periods, in more varied tasks, and with a significant "curated" component, −that is, carrying stone tools around with them and shaping stone tools for efficient carrying and safe use while cutting, and resharpening them when they became dull, damaged, or broken (Binford, 1979; Shott, 1996; Key et al., 2016).

Elsewhere (Shea, 2017a, 2017b), I have argued that hominin stoneworking follows a long-term and cumulative trend, from occasional to habitual to obligatory stone tool use. The Eastern African Plio-Pleistocene stone tool evidence supports that hypothesis' predictions for all but the later, obligatory stone tool use phase. (That phase appears to have begun after 0.2–0.3 Ma [Shea, 2017a, 2017b]).

## Discussion: Stoneworking and hominin extinctions

In a perfect world, one would be able to confidently assign specific stone tool samples to specific hominin creators. We do not live in a perfect world. Even in cases where hominin fossils identifiable to

taxon appear together with stone tools in the same sedimentary deposit, we cannot and should not assume those hominins made those stone tools, merely that fossils and stone tools were deposited between the oldest and youngest dates for that deposit. We can, however, work with FADs and LADs for hominins and for the presence/absence of Modes A-I in the Eastern African stone tool evidence. This approach is not without risks or potential problems (Bobe and Wood, 2022), but it is better than nothing. Table 2 lists the FADs and LADs for Eastern African hominin genera.

The Eastern African stone tool evidence sheds little light on australopithecines' extinction. They were around when some hominins started making stone tools between 2.5 and 3.5 Ma, but their LAD around 1.9 Ma leaves not a ripple in the stone tool evidence. If anything, evidence for stoneworking increases from that those dates onwards.

FADs for the Genus *Homo* ca. 2.3 Ma mark a major inflection point in the evidence. Around 2.5 Ma, hominins started making stone tools and littering Eastern Africa with them more regularly than previously (Key and Williams, 2026). That other parts of Africa show similar upticks in percussive stoneworking after 2.5 Ma (Plummer et al., 2025) suggests this increase in stoneworking's archaeological "visibility" was part of a wider evolutionary process and not a regional phenomenon unique to Eastern Africa.

FADs for *Homo ergaster/erectus* ca. 1.7 Ma correlate with first appearances of long core-tool (LCT) production and with the scaling-up of both hierarchical and non-hierarchical core reduction (Sharon Sharon, 2009; Sharon, 2010). The archaeological consensus holds LCTs were tools used to butcher large mammal carcasses or general purpose tools shaped for easy hand carrying (Wynn and Gowlett, 2018; Litov et al., 2026). Whatever their purposes, LCTs clearly worked well at them, for LCTs show up shortly after 1.6 Ma all over Africa and in southern Eurasia west of the Bay of Bengal and together with fossils of *Homo ergaster/erectus* and other species of the Genus *Homo*. LCTs are rarely the earliest lithic evidence for hominin activity in these regions (Shea, 2010). Instead, they seem to appear after delays of variable duration, perhaps a result of regional variation in hominins' "settling in" and augmenting improvisation with more stereotyped ways of provisioning themselves with stone tools to serve predictable needs for such artifacts.

*Paranthropus'* LAD after 1.3 Ma does not correlate with a major change in the Eastern African stone tool evidence. *Paranthropus* and *H. ergaster/erectus* were large, habitually ground-dwelling primates with similar nutritional requirements, and presumably similar ranked-choices among food sources. It is possible that more

**Table 2.** First and last appearance dates for Plio-Pleistocene hominins in Eastern Africa

| Hominins | First appearance date (FAD) | Last appearance date (LAD) |
|---|---|---|
| *Australopithecus afarensis* | 3.9 Ma (Belohdelie, Middle Awash Valley, Ethiopia). | 2.4 Ma (Bouri and Gamedah formations, Middle Awash Valley, Ethiopia). |
| *Paranthropus* spp. | 2.3 Ma (Lomekwi LO–1, Kenya). | 1.3 Ma (Bed II, Olduvai Gorge, Tanzania). |
| Early *Homo* | 2.3–2.4 Ma (A.L. 666, Hadar, Ethiopia). | 1.0–1.4 Ma (Koobi Fora Formation, Kenya, Lower Omo River Valley, Ethiopia). |
| *Homo ergaster/erectus* | 1.7 Ma (Koobi Fora & Naiyena Engol, Kenya) | 1.0 Ma (Daka, Bouri Formation, Middle Awash Valley, Ethiopia). |

habitual stone tool use or novel stone tool uses as subsistence aids among *Homo ergaster/erectus* adversely affected *Paranthropus.* If so, however, then it appears to have been a long-running process, one that played out over 600,000 years between 1.2 and 1.8 Ma.

Paranthropines were by any measure evolutionarily successful, living in a wide range of African habitats for more than two million years. Why does their extinction leave so little trace in the stone tool evidence? Though early studies cast doubt on paranthropine stone-working (Leakey, 1971), the last several decades have seen paleo-anthropologists increasingly accept the possibility that *Paranthropus* made and used stone tools (Susman, 1998; Prat, 2025). But, what if we are wrong? While *Paranthropus* might have benefitted from using stone tools for pre-oral food processing as Early *Homo* and H. *ergaster/erectus* did, making, using, and maintaining percussively-fractured stone tools could have been redundant with their evolved anatomical supports for biting and chewing tough materials. Why learn stone-working, risk injury doing so, and carry heavy stone tools around, if all they needed were the teeth already in their mouths? *Paranthropus* may have needed stone cutting tools no more so than wolves need knives or beavers need chainsaws.

## Conclusion

This paper has explored what the Eastern African Plio-Pleistocene stone tool evidence arranged in terms of Stoneworking Modes A-I tells us about hominin extinction. It tells a complex story –not a surprising finding considering the chronological scale involved and the many uncertainties about who made which stone tools, and why they did so. The strongest case for a connection between extinctions in the hominin fossil record and changes in the stone tool evidence involve Early *Homo* and *H. ergaster/erectus.* Species of the Genus *Homo* appear entangled with the stone tool evidence in ways *Paranthropus* and *Australopithecus* do not.

A broad consensus among paleontologists holds that Early *Homo*'s extinction was a pseudo-extinction ("anagenetic speciation") between at least one of those hominins and *H. ergaster/erectus.* Whether changes in stoneworking around 1.8 Ma were causes, consequences, or a bit of both in this transition invites multiple working hypotheses.

*Paranthropus*' extinction occurred after the appearance of novel features of (presumably) *H. ergaster/erectus*' habitual stoneworking around 1.8 Ma. Nevertheless, one must remain mindful of the 600,000-year offset between FADs of that stone tool evidence *Paranthropus*' LAD of 1.2 Ma. Six hundred thousand years is a lot of time. We cannot reject the hypothesis that *Paranthropus*' extinction had nothing to do with *H. ergaster/erectus*' activities.

This paper has done what it can with the Plio-Pleistocene lithic evidence from Eastern Africa in its current state. Viewing that evidence thorough the lens of Stoneworking Modes A-I does not overturn, and indeed supports, the current archaeological consensus about differences among australopithecines', paranthropines', and early *Homo*'s involvements in percussive stoneworking. This finding is remarkable, at the very least, because Stoneworking Modes A-I was not purposefully designed with that goal in mind. This suggests Stoneworking Modes A-I may have value for investigating hominin extinctions in other contexts where traditional archaeological stone tools systematics makes large-scale comparisons difficult other than in terms of "lowest common denominators." One thinks using Stoneworking Modes A-I for such purposes would be more effective and more credible if prior theory informed it, −predicting specifically how hominin extinctions should affect variability in the stone tool evidence. Archaeologists have tried to

investigate these matters by reverse-engineering "prior" theory after having documented archaeological variability for going on a century or more (Shea, 2011). A different, predictive "strategic modeling" approach (Tooby and DeVore, 1987) to the stone tool evidence will almost certainly be more useful to larger field of human origins research (Shea, 2017b).

Moving forward requires two courses of action. First, we must increase the number of Eastern African Plio-Pleistocene lithic assemblages with well-constrained ages. Second, we must test hypotheses derived from the Eastern African record with data from Southern Africa and other regions. Doing so will show us whether the Eastern African evidence is representative of continent-wide evolutionary processes or if it documents strictly regional phenomena. Organizing new evidence and previously-discovered evidence from Southern Africa (and elsewhere) in terms of Stoneworking Modes A-I will make such comparisons more convenient (or at least less fraught) than alternatives that require one to reconcile seemingly-irreconcilable conflicts between competing stone tool systematics (artifact typologies and measurement conventions) and then to re-analyze, artifact-by-artifact, the hundreds of thousands of Plio-Pleistocene stone tools reposing in museum archives. If using Stoneworking Modes A-I does not answer a particular research question, then the latter, artifact-based option remains available.

**Open peer review.** To view the open peer review materials for this article, please visit http://doi.org/10.1017/ext.2026.10014.

**Data availability statement.** All the data used in this paper are in Table 1. They are excerpted from a larger database, the Eastern African Stoneworking Survey (EAPSS) published in Shea (2020) and available in digital format at this website, https://sites.google.com/a/stonybrook.edu/john-j-shea/eastern-african-stone-tools-east-typology?authuser=0.

**Acknowledgements.** I thank my wife, Patricia Crawford, and two anonymous reviewers for their comments on a previous draft of this paper.

**Author contribution.** J.J.S. wrote the entirety of this paper himself.

**Financial support.** This work received no financial support.

**Competing interests.** The author declares no conflict of interests.

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
