## [Reviewer Report]

The author of this paper considers the relationship between the stone tool record and hominin extinctions in the African Plio-Pleistocene. They use the Stoneworking Modes A-I framework for classifying knapping evidence at various lithic sites. The last appearance dates for several hominin genera (Australopithecus, Paranthropus, and Homo) are compared to the changes in knapping behaviour. The author concludes that significant change occurs in the stone tool record with the appearance of LCTs c. 1.7 Ma, coinciding best with the last appearance of early Homo and the first appearance of Homo erectus/ergaster. The loss of Australopithecus and Paranthropus from the fossil record do not appear to correlate with any significant change in knapping behaviour.

This paper does well to combine several streams of evidence into a convincing argument for the relevance of stone tools in the Homo narrative. The use of the Stoneworking Modes presents a refreshing analysis of lithic sites based on behaviours/skills demonstrated, rather than focussing on applying potentially loaded technological labels. The figure (1) explaining these modes is clear and will be particularly useful for non-lithic specialists. The main conclusion of this paper (that the genus Homo appears to be the most intertwined with the Plio-Pleistocene stone tool record) is reasonable considering the apparent coincidence between the FAD of H. erectus/ergaster and the appearance of more complex knapping behaviour including LCTs. Overall, this paper provides an interesting perspective on the long-lived debate over which hominin species/genera made and used stone tools.

A few minor suggestions:

Line 68: “motives for creating stone cutting tools.” - makes the sentence clearer

Line 80: “and footprints attributed to them” - typo

Line 100: superscript for cm3

Line 112: “wear-trace analyses hint” - typo

Line 145: “This database is updated annually and is available online at” - typo

Line 172: “artifacts include backed and/or truncated” - typo

Lines 225-227: this section suggests a gap in known stone tool sites between LOM-3 and c. 2.6 Ma, however there are two sites that fall in this gap that haven’t been considered here: Nyayanga and Namorotukunan. They should be added for completeness/omission justified, although I doubt they will alter the conclusions.

Line 295: “the stones remain silent” - typo

Line 307: “doubt on whether paranthropine stoneworking...” - seems to be part of the sentence missing here

Lines 304-315: suggests Paranthropus may not have needed stone tools because of their dental adaptations – this seems to contradict the opening statement that their lack of honing canines suggests a motive for needing to use stone tools

Line 328: “in this transition remains a question that invites” - typo

Lines 330-337: this passage attempts to link the extinction of Paranthropus to the advent of increased behavioural complexity seen after the appearance of H. erectus/ergaster. But, given the large amount of time between this and the eventual apparent extinction of Paranthropus, it may also be possible that some other external, non-Homo related factor contributed to its extinction.

Lines 332-333: “of that stone tool evidence and Paranthropus’ LAD” - typo

Data Availability Statement: should this refer to table 1, not table 2?

Text flips between using “we” and “I”, and flips between italicising Paranthropus and not (Homo is always italicised).

I would suggest the following changes to Table 2:

Australopithecus LAD:

- Eastern Africa = c. 2.4 Ma (Middle Awash, Ethiopia [Bouri and Gamedah]).

- Southern Africa = could be as young as 1.5 Ma (Malapa Facies F)

Paranthropus:

The description of Paranthropus (line 82) includes aethiopicus, but the FAD/LADs in Table 2 don’t seem to incorporate this.

- FAD (with aethiopicus) = could be as early as c. 3 Ma (Lomekwi LO-1)

- FAD (with boisei) = 2.7 Ma (Lomekwi LO-3)

- LAD in eastern Africa = c. 1.3 Ma (Olduvai Gorge)

- LAD in southern Africa = c. 1 Ma (Swartkrans member 3).

Early Homo (habilis + rudolfensis):

- FAD = c. 2.35 Ma (A.L. 666) but could be as early as 3.25 Ma (KNM-ER 2603 – just tooth fragments) or c. 2.7-2.8 Ma (Ledi-Geraru) if including early Homo indet. fossils.

- LAD = c. 1.4 Ma (Okote member at Koobi Fora and Ileret) but could be as young as 1 Ma (OMO 7-69-19).

Homo erectus/ergaster:

- FAD in eastern Africa = c. 1.7 Ma (Koobi Fora and Naiyena Engol)

- FAD in southern Africa = c. 2 Ma (Drimolen and Sterkfontein).

Considering the revised FAD and LAD dates for Table 2, the statements (lines 291-292 + 325-329) regarding anagenetic speciation from Homo habilis to H. erectus/ergaster seem unsupported given there is an overlap of at least 300 ka, possibly as much as 700ka, between the two. This, of course, does not rule out the possibility that H. erectus/ergaster did evolve from H. habilis, just not in an anagenetic fashion.

Following this, I would also suggest caveating the conclusions with a consideration of the disadvantages of using fossil dates as absolute FADs and LADs compared to potential actual FADs and LADs (e.g., Bobe, R. and Wood, B.A. 2022. Estimating origination times from the early hominin fossil record. Evolutionary Anthropology: Issues, News, and Reviews, 31(2), pp.92-102).

---

## [Reviewer Report]

Major Comments

This paper seeks to use the authors new lithic classification system or stoneworking modes to examine hominin extinctions between 3.5-0.8Ma in East Africa. The author reports, in essence negative results for two of the species examined. For both Australopithecus and Paranthropus there is no discernible trace in the stoneworking modes present in the associated archaeological record. It is noted that there is a clear shifting stoneworking modes, in terms of more complex reduction strategies and an increase in retouched forms at around 1.9Ma, broadly contemporaneous with the emergence of Homo ergaster / erectus, as well as correlating with the last appearance of early Homo. The paper ends with a call for more well dated Plio-Pleistocene sites as well as expanding technological analysis to early assemblages in South Africa to test against the patterns observed in East Africa.

This paper presents an interesting topic, and I view it more in the category of a broad review paper as opposed to a paper where a specific hypothesis is being tested using new data or a new method. Having said, this however, it was somewhat unclear in the introduction what the main aim of the paper was, and I would suggest a short bridging paragraph at the end of the introduction to present the main question being addressed.

While I commend the author’s goal of promoting a more systematic and technologically driven approach to East African lithic assemblages (Shea, 2020), I remain unconvinced that applying this method yields a genuinely new perspective on the hominin technological attributions. As I understand it, the stoneworking A-I framework classifies lithic assemblages by their technological attributes. This approach is presented as a way to overcome the longstanding reliance on typological classification, which has been prevalent historically in East African lithic analyses (Leakey, 1971), and to avoid complications of named industries such as the Oldowan Developed Oldowan A (DOA), Developed Oldowan B (DOB) and Acheulean (but see (Semaw et al., 2009; de la Torre and Mora) for a discussion into the validity of the DOA and DOB). However, similar technological frameworks have been used in East African archaeological research for some time, largely derived from European (particularly French and Spanish) approaches to technological lithic analysis (Mora et al., 1992; de la Torre et al., 2003; de la Torre, 2004; Delagnes and Roche, 2005; de la Torre and Mora, 2005; Diez-Martin et al., 2010; Stout et al., 2010; Roche et al., 2018; Mesfin and Texier, 2022; Mesfin et al., 2026). Many of these employ comparable schematic systems for describing Plio-Pleistocene core exploitation. Likewise, the technological classification of retouched pieces adopted in this manuscript is very similar to the technological framework for retouch analysis proposed by Laplace (Laplace, 1974).

It has long been accepted that the emergence of Acheulean technology coincides with the appearance of Homo ergaster / erectus as well as the disappearance of early Homo (de la Torre, 2011; De La Torre, 2016; de la Torre et al., 2021). There do, however, also seems to be nuances to this transition that including retouched pieces as a defining feature of Homo ergaster / erectus perhaps glosses over (de la Torre and Mora, 2018; de la Torre et al., 2021). Specifically, it is apparent that at a number of Oldowan sites retouched pieces are present and in some cases may hint at more complex secondary tool use by early Homo (de la Torre et al., 2021).

Furthermore, there are tentative indications that Parathropus may be associated with early Oldowan assemblages (Plummer et al., 2023), and may have use the same technology as early Homo. If so, their extinction their extinction would leave no distinct archaeological signature, given that Oldowan technology was also employed by early Homo. This manuscript might benefit form a more in-depth discussion of these nuances in the Plio-pleistocene archaeological record.

Another point that could benefit from some clarification is the statement in the Results section:

“The stoneworking methods present before 1.9 Ma are, to put it simply, simple. They work on pretty much any rock with even an approximation of conchoidal fracture. That they appear among artifacts made by novice craft/hobby knappers and even persons with little or no prior experience or knowledge about stone tools”.

This characterisation oversimplifies the extensive techno-typological literature on Oldowan assemblages. While these assemblages are undoubtedly more simple than Middle or Upper Palaeolithic technologies or finely crafted Acheulean handaxes, many, nonetheless display a considerable degree of technical knowledge and planning. At Lokalalei 2C, for instance, cores were reduced through extensive, systematic flaking sequences, in some cases involving more than 70 removals (Delagnes and Roche, 2005). Similarly, the Omo assemblages (de la Torre, 2004) show knappers deploying considerable skill to apply complex reduction strategies on small quartzite cobbles, maximising flake output from challenging raw materials. To assert, without engaging with the primary literature, that such assemblages could be produced by individuals with ‘no prior experience or knowledge about stone tools’ is difficult to support.

Furthermore, the key experimental study cited in support of this claim (Snyder et al. 2022) has notable limitations: it employed naïve modern human subjects working with standardised glass cores with uniformly acute angles, presents only illustrated reconstructions of select examples rather than photographic documentation of the cores themselves, and lacks direct comparison with the archaeological record. In doing so, it risks underestimating the genuine complexity evident within Oldowan assemblages, where knappers were likely making deliberate technological decisions.

Perhaps another area of oversimplification concerns the classification of spheroids and subspheroids under the broad umbrella of anvil percussion. A substantial body of literature has debated the extent to which these artefacts represent intentional shaping versus incidental by-products of other reduction activities (Sahnouni et al., 1997; Mora R, 2005; Texier and Roche, 2014; Cabanès et al., 2022; Barsky et al., 2025) This ambiguity is particularly acute for subspheroids from East African contexts, many of which exhibit evidence of multiple, overlapping reduction histories, combining freehand percussion, bipolar exploitation, and extensive subsequent percussion, suggesting they passed through several distinct use episodes and technological tasks rather than resulting from a single, discrete anvil percussion event (Mora and de la Torre, 2005a; de la Torre and Mora, 2018).

It also, seems that a large portion of the manuscript describes the stoneworking modes themselves rather than interrogating why those extinctions left no lithic signal. There may be several competing explanations for this, which could be further explored in the text, such as dietary niche separation, habitat differences, demographic collapse unrelated to tool use, climate.

It might also be interesting to explore whether the inflection points that the authors notes, at 3.4Ma and 1.9Ma are genuine evolutionary signals or might partly be explained by biases in where archaeologists have conducted long term research. There could be a short discussion of the potential biases that may be inherent in the stoneworking modes, such as sampling bias, the geographical coverage, or how representative sites are distributed across time and space.

A more general issue I would like to constructively raise regarding the manuscript is that there seems to be a general lack of citations for statements in the text. I have listed below the instances where I felt that a citation would strengthen the statement being made.

Minor Comments

Line 48: Change “stalks the…”

Line 67: “Miocene ancestors’ large self-sharpening canine teeth” Reference needed here.

Line 68: “…and so all of these hominins had motives for stone cutting tool.” Reference needed here. I am not sure one can be confident of hominin motivations.

Para starting on line 71: This paragraph could do with references to the primary literature on hominin sexual dimorphism.

Para starting on line 76: The statements in this paragraph and data on hominin stature and cranial capacity all need references.

Line 80: “Their post-cranial remains and footprints attribute to them at Laetoli in Tanzania show they walked bipedally, but with some differences from living human bipedalism.” Reference needed here. Also there is a typo “attributed”.

Para starting on line 82: Same comment as above. Primary references should be given for all statements.

Line 92: References are needed here, especially if the author is going to state that views diverge.

Two paragraphs starting on line 95 ending on lone 105: There are multiple statements regarding the vital statistics of early Homo and Homo ergaster/erectus as well as their morphologies that require citations to the primary literature where these data can be found.

Line 115: For clarity might I suggest the following “with the stone tools that recent humans use.....”

Line 118: “…traces of extensive percussion that did not result in large-scale fracturing but rather “comminution” (multiple overlapping and incompletely-propagated fractures)(Diez-Martín et al. 2009)” This is not the primary reference for this type of percussive artefact, please see (de la Torre and Mora, 2005; Mora and de la Torre, 2005b)

Line120: “Indeed, we should not reject the hypothesis that noise-making for some social and/or signaling purpose may have been involved.”

This seems like an untestable hypothesis, and as such should be referred to as a suggestion as opposed to a hypothesis. As it stands its more of a statement of possibility. I am unclear about how this could be tested using the available archaeological evidence.

Line 124: This is not the correct reference for these terms. Please include the primary references for the Lomekwian (Harmand et al., 2015), Pre-Oldowan (Roche, 1988), Oldowan (Leakey, 1971), Developed Oldowan (Leakey, 1971) and Early Acheuelan

Line 125: “Archaeological views diverge over by which criteria to assign stone tool assemblages to one or the other of these industries, these industries’ historical relationships to one another, and to which hominin(s) to attribute each of them.”

This statement could benefit from some references to this divergence of views.

Line 155: Anvil percussion here, presumably refers to passive hammer knapping (as in Lomekwi 3). In this case, would not also a diagnostic artifact be the large unidirectional cores, with percussive damage and small edge opposed scars along the platform edge (Harmand et al., 2015). An artefact type that is not found in the proceeding Oldowan assemblages.

Line 184: The use of the phrase “long core-tool (LCT)” is somewhat confusing here. This is not the standard definition of LCT, which typically referees to Large Cutting Tools. If you are using a different terminology here, please also note that LCT is commonly used as the acronym for Large Cutting Tool so as to avoid confusion to the reader.

Line 221: “Of course, some teaching and learning….” This statement hides quite a substantial body of recent literature regarding Plio-Pleistocene social learning. It is unclear whether it would, of course, be the case that teaching and learning took place during this time. Perhaps a citation or two regarding social learning capacities during the Plio-Pleistocene is needed here.

Line 226: “…nearly a million years long between the Lomekwi 3 evidence and the next dated appearances of lithic artifacts around 2.5-2.6 Ma.” Please see the recent papers by (Plummer et al., 2023; Braun et al., 2025).

Lines 228: reference needed for “Even after stoneworking seems to begin in earnest after 2.5 Ma…”

Line 234: “Modes D, E, and F all require more time….” Please see the recent paper by (Key et al., 2026)

Line 240: “….creates sinuous edges that cut poorly.” Refence needed for this statement.

Line 252: “…carrying stone tools around with them and shaping stone tools for efficient carrying.” A reference for intentionally shaping Acheulean handaxes to be efficiently carried would be required here.

Line 341: Please see the recent technological analysis by (Mesfin et al., 2026).

References

Barsky, D., Sala-Ramos, R., Bargalló, A., Muller, A., Sharon, G., Grosman, L., 2025. Identifying a Knapping Signature for Lower Paleolithic Spheroid Reduction. Journal of Paleolithic Archaeology. 8, 29.

Braun, D.R., Palcu Rolier, D.V., Advokaat, E.L., Archer, W., Baraki, N.G., Biernat, M.D., Beaudoin, E., Behrensmeyer, A.K., Bobe, R., Elmes, K., Forrest, F., Hammond, A.S., Jovane, L., Kinyanjui, R.N., de Martini, A.P., Mason, P.R.D., McGrosky, A., Munga, J., Ndiema, E.K., Patterson, D.B., Reeves, J.S., Roman, D.C., Sier, M.J., Srivastava, P., Tuosto, K., Uno, K.T., Villaseñor, A., Wynn, J.G., Harris, J.W.K., Carvalho, S., 2025. Early Oldowan technology thrived during Pliocene environmental change in the Turkana Basin, Kenya. Nature Communications. 16, 9401.

Cabanès, J., Borel, A., Preysler, J.B., Lourdeau, A., Moncel, M.-H., 2022. Palaeolithic polyhedrons, spheroids and bolas over time and space. PLOS ONE. 17, e0272135.

de la Torre, I., 2004. Omo revisited: Evaluating the technological skills of pliocene hominids. Current Anthropology. 45, 439–465.

de la Torre, I., 2011. The Early Stone Age lithic assemblages of Gadeb (Ethiopia) and the Developed Oldowan/early Acheulean in East Africa. Journal of Human Evolution. 60, 768–812.

De La Torre, I., 2016. The origins of the Acheulean: past and present perspectives on a major transition in human evolution. Philosophical Transactions of the Royal Society B: Biological Sciences. 371, 20150245.

de la Torre, I., Benito-Calvo, A., Martín-Ramos, C., McHenry, L.J., Mora, R., Njau, J.K., Pante, M.C., Stanistreet, I.G., Stollhofen, H., 2021. New excavations in the MNK Skull site, and the last appearance of the Oldowan and Homo habilis at Olduvai Gorge, Tanzania. Journal of Anthropological Archaeology. 61, 101255.

de la Torre, I., Mora, R., 2005. Technological strategies in the lower Pleistocene at Olduvai Beds I & II. Etudes et Recherches Archeologiques de l’Universite de Liege, Liege.

de la Torre, I., Mora, R., 2014. The Transition to the Acheulean in East Africa: an Assessment of Paradigms and Evidence from Olduvai Gorge (Tanzania). Journal of Archaeological Method and Theory. 21, 781–823.

de la Torre, I., Mora, R., 2018. Oldowan technological behaviour at HWK EE (Olduvai Gorge, Tanzania). Journal of Human Evolution, From the Oldowan to the Acheulean at Olduvai Gorge (Tanzania). 120, 236–273.

de la Torre, I., Mora, R., Domínguez-Rodrigo, M., de Luque, L., Alcalá, L., 2003. The Oldowan industry of Peninj and its bearing on the reconstruction of the technological skills of LowerPleistocene hominids. Journal of Human Evolution. 44, 203–224.

Delagnes, A., Roche, H., 2005. Late Pliocene hominid knapping skills: The case of Lokalalei 2C, West Turkana, Kenya. Journal of Human Evolution. 48, 435–472.

Diez-Martin, F., Sanchez Yustos, P., Domínguez-Rodrigo, M., Mabulla, A.Z.P., Bunn, H.T., Ashley, G.M., Barba, R., Baquedano, E., 2010. New insights into hominin lithic activities at FLK North Bed I, Olduvai Gorge, Tanzania. Quaternary Research. 74, 376–387.

Harmand, S., Lewis, J.E., Feibel, C.S., Lepre, C.J., Prat, S., Lenoble, A., Boës, X., Quinn, R.L., Brenet, M., Arroyo, A., 2015. 3.3-million-year-old stone tools from Lomekwi 3, West Turkana, Kenya. Nature. 521, 310–315.

Key, A., Stileman, F., Fedato, A., Eren, M., 2026. Acheulean Expediency Potential: Handaxe Manufacturing Time Costs, Covariates and Skill. Lithic Technology. 0, 1–25.

Laplace, G., 1974. La Typologie Analytique et Structurale: base rationnelle d’etude des industries lithiques et osseuses. In: Banques de Donnees Archeologiques, Colloques Nationaux C.N.R.S. Marseille, Paris, pp. 94–143.

Leakey, M.D., 1971. Olduvai Gorge: Volume 3, Excavations in Beds I and II, 1960-1963. Cambridge University Press.

Mesfin, I., Caruana, M.V., Kuman, K., 2026. Application of the Chaîne Opératoire and Techno-functional Approaches To Oldowan Cores and Retouched Pieces from Swartkrans and Sterkfontein, South Africa. Journal of Paleolithic Archaeology. 9, 11.

Mesfin, I., Texier, P.-J., 2022. Prepared core technology from the Early Pleistocene site of Nyabusosi 18, Uganda. Journal of Archaeological Science: Reports. 46, 103695.

Mora R, de la T.I., 2005. Percussion tools in Olduvai Beds I and II (Tanzania): Implications for early human activities. Journal of Anthropological Archaeology. 24, 179–192.

Mora, R., de la Torre, I., 2005a. Percussion tools in Olduvai Beds I and II (Tanzania): Implications for early human activities. Journal of Anthropological Archaeology. 24, 179–192.

Mora, R., de la Torre, I., 2005b. Percussion tools in Olduvai Beds I and II (Tanzania): Implications for early human activities. Journal of Anthropological Archaeology. 24, 179–192.

Mora, R., Moreno, J.M., Batlle, X.T., 1992. Un proyecto de análisis: el Sistema Lógico Analítico (SLA). Treballs d’Arqueologia. 173–199.

Plummer, T.W., Oliver, J.S., Finestone, E.M., Ditchfield, P.W., Bishop, L.C., Blumenthal, S.A., Lemorini, C., Caricola, I., Bailey, S.E., Herries, A.I.R., Parkinson, J.A., Whitfield, E., Hertel, F., Kinyanjui, R.N., Vincent, T.H., Li, Y., Louys, J., Frost, S.R., Braun, D.R., Reeves, J.S., Early, E.D.G., Onyango, B., Lamela-Lopez, R., Forrest, F.L., He, H., Lane, T.P., Frouin, M., Nomade, S., Wilson, E.P., Bartilol, S.K., Rotich, N.K., Potts, R., 2023. Expanded geographic distribution and dietary strategies of the earliest Oldowan hominins and Paranthropus. Science. 379, 561–566.

Roche, H., 1988. Technological evolution in early hominids. In: The Wenner-Gren Foundation Symposium“ Behavior Hominids”. pp. 97–98.

Roche, H., de la Torre, I., Arroyo, A., Brugal, J.-P., Harmand, S., 2018. Naiyena Engol 2 (West Turkana, Kenya): a Case Study on Variability in the Oldowan. African Archaeological Review. 35, 57–85.

Sahnouni, M., Schick, K., Toth, N., 1997. An Experimental Investigation into the Nature of Faceted Limestone “Spheroids” in the Early Palaeolithic. Journal of Archaeological Science. 24, 701–713.

Semaw, S., Rogers, M., Stout, D., 2009. The Oldowan-Acheulian Transition: Is there a “Developed Oldowan” Artifact Tradition? 173–193.

Shea, J.J., 2020. Prehistoric stone tools of eastern Africa: A guide. Cambridge University Press.

Stout, D., Semaw, S., Rogers, M.J., Cauche, D., 2010. Technological variation in the earliest Oldowan from Gona, Afar, Ethiopia. Journal of Human Evolution. 58, 474–91.

Texier, P.-J., Roche, H., 2014. Polyèdre, sub-sphéroïde, sphéroïde et bola: des segments plus ou moins longs d’une même chaîne opératoire. Cahier noir. 31–40.

---

## [Editor Report]

Thanks are extended to the author for their novel and interesting submission. The manuscript is somewhere between a review and a data-led analysis, and the two reviewers were very positive about the manuscript’s aims and intent. A perspective I share. It is a straightforward ‘revise’ recommendation. Both reviewers provide very detailed comments which are helpful and constructive. R2 provides a few more major comments which I would agree are important to address, hence the recommendation, but this can be easily done and should not take too long.

---

## [Editor Report]

Thanks are extended to the author for their revised manuscript and response to the reviewers. I am pleased to say that subject to a few minor changes caught during my reading of the revised manuscript, I consider it suitable for publication.

- This is not a required revision, but revisiting the new text at the end of the introduction may strengthen your argument for using modes A-I. An invitation to use them is not necessarily justification for use (albeit worth noting given the special issue), and I suspect the reviewer’s comment was requesting a little more methodological reasoning for why modes A-I are providing something new. You do this very well in the ‘response to the reviewers’ letter, and I suspect some of this could be repeated. As you say, application of modes A-I is complementary to alternative approaches, not contradictory, and its use tests insight gained via other means and strengthens the field through reproducibility and retesting. This is an admirable aim of the scientific approach and, as you say, modes A-I can help to remove uncertainties between analysts, increase ease of use and allow more inclusive comparisons. I appreciate you touch on these aspects elsewhere in the text, but if you do want to justify the use of mode A-I at the end of the introduction, touching on these convincing arguments would strengthen your case to the reader from the outset (there may be some balancing needed relative to later comments on the method). 

Final typos:

- ‘Stone tools are essentially artificial teeth (Shea 2017b), and so, all of these hominins had possible motives for using stone cutting tools, either naturally-fractured rocks (Eren et al., 2025) or edges of broken bone.’ – Do you specifically mean fossilized bone (and therefore stone)? If not, a little adjusting of the sentence to make sense of the inclusion of ‘bone’ could help (given the first half of the sentence explicitly focuses on stone). 

- ‘…all appear to have had opposable thumbs and precision grasping…’ – Citing Kivell et al. (2023, AJBA) would suitably cover this information. Perhaps add ‘…precision grasping, albeit with some variation in capability’ as a caveat to the variation present in the manual fossil record. 

- Line 195: ‘… from roughly clasts…’ – check phrasing. 

- Line 285: ‘Carefully retouching the edge of a flake carefully…’. – Double use of ‘carefully’. Check phrasing. 

- Line 289: ‘Striking a both dorsal and ventral sides…’. Check phrasing. Cut ‘a’?

- Line 323: ‘…perfect world, one be able to…’ Check phrasing. Add ‘would’?

- Line 387: ‘…story, -unsurprising…’. Check punctuation.

---

## [Editor Report]

Thanks are extended to the author for returning their revised submission. Although the tracked changes document highlights a large number of changes to the text, the majority detail prior reviewed-text being moved from one position to another. Other additions and changes are small; principally structural changes within sentences, alongside a few new sentences addressing my response to the last version of the manuscript. I am happy to recommend acceptance.